# Acute Hyperglycemia Aggravates Lung Injury via Activation of the SGK1–NKCC1 Pathway

**DOI:** 10.3390/ijms21134803

**Published:** 2020-07-07

**Authors:** Chin-Pyng Wu, Kun-Lun Huang, Chung-Kan Peng, Chou-Chin Lan

**Affiliations:** 1Department of Critical Care Medicine, Landseed International Hospital, Tao-Yuan 32449, Taiwan; wucp@landseed.com.tw; 2Division of Pulmonary Medicine, National Defense Medical Center, Tri-Service General Hospital, Taipei 11490, Taiwan; kun@ndmctsgh.edu.tw (K.-L.H.); kanpeng@mail.ndmctsgh.edu.tw (C.-K.P.); 3Institute of Undersea and Hyperbaric Medicine, National Defense Medical Center, Taipei 11490, Taiwan; 4Division of Pulmonary Medicine, Taipei Tzu Chi Hospital, Buddhist Tzu Chi Medical Foundation, New Taipei City 23142, Taiwan; 5School of Medicine, Tzu-Chi University, Hualien 97004, Taiwan

**Keywords:** acute lung injury, sodium-potassium-chloride co-transporter isoform 1, hyperglycemia, serum-glucocorticoid kinase 1

## Abstract

Acute lung injury (ALI) is characterized by severe hypoxemia and has significantly high mortality rates. Acute hyperglycemia occurs in patients with conditions such as sepsis or trauma, among others, and it results in aggravated inflammation and induces damage in patients with ALI. Regulation of alveolar fluid is essential for the development and resolution of pulmonary edema in lung injury. Pulmonary sodium-potassium-chloride co-transporter 1 (NKCC1) regulates the net influx of ions and water into alveolar cells. The activation of with-no-lysine kinase 4 (WNK4), STE20/SPS1-related proline/alanine rich kinase (SPAK) and the NKCC1 pathway lead to an increase in the expression of NKCC1 and aggravation of ALI. Moreover, hyperglycemia is known to induce NKCC1 expression via the activation of the serum-glucocorticoid kinase 1 (SGK1)–NKCC1 pathway. We aim to evaluate the influence of acute hyperglycemia on the SGK1–NKCC1 pathway in ALI. ALI was induced using a high tidal volume for four hours in a rat model. Acute hyperglycemia was induced by injection with 0.5 mL of 40% glucose solution followed by continuous infusion at 2 mL/h. The animals were divided into sham, sham+ hyperglycemia, ALI, ALI + hyperglycemia, ALI + inhaled bumetanide (NKCC1 inhibitor) pretreatment, ALI + hyperglycemia + inhalational bumetanide pretreatment, and ALI + hyperglycemia + post-ALI inhalational bumetanide groups. Severe lung injury along with pulmonary edema, alveolar protein leakage, and lung inflammation was observed in ALI with hyperglycemia than in ALI without hyperglycemia. This was concurrent with the higher expression of pro-inflammatory cytokines, infiltration of neutrophils and alveolar macrophages (AM) 1, and NKCC1 expression. Inhalational NKCC1 inhibitor significantly inhibited the SGK1–NKCC1, and WNK4–SPAK–NKCC1 pathways. Additionally, it reduced pulmonary edema, inflammation, levels of pro-inflammatory cytokines, neutrophils and AM1 and increased AM2. Therefore, acute hyperglycemia aggravates lung injury via the further activation of the SGK1–NKCC1 pathway. The NKCC1 inhibitor can effectively attenuate lung injury aggravated by acute hyperglycemia.

## 1. Introduction

Acute lung injury (ALI) is characterized by poor pulmonary gas exchange, hypoxemia, and frequently results in significant morbidity and mortality in intensive care units (ICUs) [1]. Even with modern treatment strategies for ALI, certain challenges remain in the treatment of patients with ALI. Therefore, it is important to elucidate the pathophysiology of ALI and attempt to develop possible effective treatment modalities [2].

The pathophysiological hallmarks of ALI are acute inflammation, pulmonary vascular hyper-permeability, and pulmonary edema [2]. Alveolar flooding is one of the notable features of ALI. Regulation of alveolar fluid is crucial, as the impaired clearance of alveolar fluid plays a critical role in ALI. The sodium-potassium-chloride co-transporter (NKCC) in lung epithelial cells regulates alveolar fluid transport by coupling the transportation of sodium (Na^+^), chloride (Cl^−^), and potassium (K^+^) [3]. Although two isoforms of NKCC are present in the body, only NKCC1 is expressed in lungs [3]. 

STE20/SPS1-related proline/alanine rich kinase (SPAK) is a downstream substrate of with-no-lysine kinase 4 (WNK4) and upstream regulators of NKCC1 [4]. The activity of NKCC1 is regulated by a signaling cascade of WNK4 and SPAK. The activation of the WNK4–SPAK pathway can enhance the expression of NKCC1. High NKCC1 expression results in water influx in the alveoli and induces alveolar edema [3]. Therefore, the WNK4–SPAK–NKCC1 pathway plays a critical role in the regulation of alveolar fluid clearance in ALI. Lin et al. reported that the increased expression of phosphorylated NKCC1 increases the severity of hyperoxia-related ALI [4]. Additionally, we previously demonstrated that enhanced NKCC1 expression induces the aggravation of ischemia reperfusion-induced ALI [5]. In these two studies, the manipulations for inhibiting NKCC1 expression could reduce the severity of lung injury [4,5].

Acute hyperglycemia often occurs in patients with severe sepsis, trauma, and other systemic inflammatory response syndromes [6]. A previous study reported that approximately 12% of hyperglycemic patients in ICUs had no history of diagnosis of diabetes [7]. Acute hyperglycemia in patients either with or without diabetes is suggested to be associated with poorer outcomes [6]. Acute hyperglycemic patients in ICUs exhibited a three-fold mortality rate compared to normoglycemic patients [7]. Acute hyperglycemia caused by the aggravation of inflammatory mediators is known to serve as a factor in exacerbating lung injury [8]. Aljada et al. reported that acute hyperglycemia leads to aggravated lung inflammation [8]. Lapar et al. showed that acute hyperglycemia could aggravate lung injury in ALI with elevated expression of pro-inflammatory cytokines and increased lung inflammation [9]. These studies suggest that acute hyperglycemia aggravates inflammation and damage in ALI.

The alveolar macrophages (AM) are one of the major inflammatory cells in the lungs and they aid in the maintenance of immunological homeostasis in the lungs [10]. AMs can differentiate into classically activated macrophages (AM1) and alternatively activated macrophages (AM2) [10]. AM1 secretes pro-inflammatory cytokines, such as tumor necrosis factor-α (TNF-α), interleukin-1β (IL-1β), IL-6, and IL-8, which further induce lung damage [10]. AM2 produces anti-inflammatory cytokines (such as IL-4, IL-10, and IL-13) and can accelerate the resolution of inflammation after ALI [10]. Under usual physiological conditions, there is a balance between the levels of AM1 and AM2 in the lungs. In ALI, AM1 accumulation at the site of injury is an early response to lung injury and AM1 triggers the release of pro-inflammatory cytokines, which leads to lung inflammation and damage [10]. Herein, we also investigated the effects of hyperglycemia and NKCC1 inhibitor treatment on AMs in ALI. 

The mechanisms underlying acute hyperglycemia in ALI are not yet completely understood. Previous research on the effects of acute hyperglycemia on the expression of pulmonary NKCC1 and alveolar fluid regulation is lacking. However, the expression of serum-glucocorticoid kinase 1 (SGK1) is known to be upregulated in hyperglycemia [11], and increased SGK1 expression is known to augment the expression of NKCC1 [12]. Since NKCC1 plays an important role in the regulation of alveolar water levels, hyperglycemia may potentially aggravate lung injury by activating the SGK1–NKCC1 pathway. 

This study aimed to evaluate: (1) the effects of acute hyperglycemia on lung inflammation and edema in ALI, (2) the effects of acute hyperglycemia on the SGK1–NKCC1 pathways in ALI, and (3) the therapeutic effects of the NKCC1 inhibitor in lung injury with acute hyperglycemia. Understanding the mechanisms underlying the aggravation of lung injury in acute hyperglycemia may aid the development of therapeutic strategies.

## 2. Results

### 2.1. Blood Glucose Levels, Oxygenation, and Dynamic Compliance of Lungs (Cdyn)

During the four-hour duration of the experiment, the blood glucose levels (Figure 1A) in the rats of sham + hyperglycemia (HG), ALI + HG, ALI + HG + bumetanide pretreatment (pre-B), and ALI + HG + post-ALI bumetanide (post-B) groups were 400–500 mg/dL, which was significantly higher than the blood glucose levels in the rats of sham, ALI, and ALI + pre-B groups (*p* < 0.05). At the fourth hour, the partial pressure of oxygen in the arterial blood/ inspired oxygen fraction (PaO_2_/FiO_2_) ratios (Figure 1B) reduced significantly in rats with ALI or ALI + HG than in rats of the sham group (*p* < 0.05). However, the PaO_2_/FiO_2_ was lower in rats with ALI + HG than in those with ALI (*p* < 0.05). The rats that received pretreatment or post-ALI treatment with the NKCC1 inhibitor had higher PaO_2_/FiO_2_ ratios than those without NKCC1 inhibitor (*p* < 0.05, comparing ALI vs. ALI + pre-B, ALI + HG vs. ALI + HG + pre-B, and ALI + HG vs. ALI + HG + post-B groups). At the fourth hour, the Cdyn value (Figure 1C) reduced significantly in rats with ALI or ALI + HG (*p* < 0.05) than in those from the sham group. The Cdyn value was lower in rats with ALI + HG than in rats with ALI (*p* < 0.05). Rats that underwent pretreatment or post-ALI treatment with the NKCC1 inhibitor had higher Cdyn value than those without the NKCC1 inhibitor (*p* < 0.05, comparing ALI vs. ALI + pre-B, ALI + HG vs. ALI + HG + pre-B or post-B groups).

### 2.2. Effects of Hyperglycemia and NKCC1 Inhibitor Treatment on Pulmonary Edema and Alveolar Protein Leakage

The lung weight/body weight (LW/BW) (Figure 2A) and the lung wet/dry (W/D) ratios (Figure 2B) were significantly higher in rats with ALI (*p* < 0.05) than in rats of the sham group. However, the LW/BW and W/D ratios were significantly higher in rats of ALI + HG than in those of ALI (*p* < 0.05, comparing ALI and ALI + HG). The pretreatment or post-ALI treatment with the NKCC1 inhibitor reduced pulmonary edema in groups with ALI and ALI + HG (*p* < 0.05, comparing ALI vs. ALI + pre-B, ALI + HG vs. ALI + HG + pre-B, and ALI + HG vs. ALI + HG + post-B groups).

Alveolar protein leakage (Figure 2C) was prominent in rats with ALI and ALI + HG (*p* < 0.05). However, the total protein (TP) concentration in BALF was significantly higher in the ALI + HG group than in the ALI group (*p* < 0.05). Pretreatment or post-ALI treatment with the NKCC1 inhibitor led to a reduction in the concentration of TP in rats from the ALI and ALI + HG groups (*p* < 0.05, comparing ALI vs. ALI + pre-B, ALI + HG vs. ALI +HG + pre-B or post-B groups).

### 2.3. Effects of Hyperglycemia and NKCC1 Inhibitor Treatment on Lung Inflammation and Neutrophilic Infiltration

Rats from the sham (Figure 3A) and sham + HG (Figure 3B) groups exhibited normal histology. Sections derived from the rats of the ALI (Figure 3C) and ALI + HG (Figure 3D) groups showed signs of severe lung injury with prominent alveolar exudation, alveolar septum thickening, and neutrophil sequestration. Treatment with the NKCC1 inhibitor led to a reduction in lung injury, which was demonstrated in the ALI + pre-B (Figure 3E), ALI + HG + pre-B (Figure 3F), and ALI + HG + post-B groups (Figure 3G).

The neutrophilic counts (Figure 3H) were higher in rats with ALI and ALI + HG (*p* < 0.05, compared to that in the sham group). However, the neutrophilic counts were significantly higher in the ALI + HG group than in the ALI group (*p* < 0.05). Pretreatment or post-ALI treatment with the NKCC1 inhibitor led to a reduction in the neutrophilic counts in rats from ALI and ALI + HG groups (*p* < 0.05, comparing ALI vs. ALI + pre-B, ALI + HG vs. ALI + HG + pre-B or post-B groups). The ALI scores (Figure 3I) were higher in rats with ALI (*p* < 0.05, compared to that in the sham group) and further increased in ALI +HG (*p* < 0.05, compared to that in the ALI group). Pretreatment or post-ALI treatment with the NKCC1 inhibitor led to a decrease in ALI scores in rats from ALI and ALI + HG groups (*p* < 0.05, comparing ALI vs. ALI + pre-B, ALI + HG vs. ALI + HG + pre-B or post-B groups).

### 2.4. Effects of Hyperglycemia and NKCC1 Inhibitor Treatment on the Levels of Cytokines

Higher levels of serum TNF-α (Figure 4A), IL-1β (Figure 4B), cytokine-induced neutrophil chemoattractant 1 (CINC-1, Figure 4C) and lower level of IL-10 (Figure 4D) were observed in rats from the ALI and ALI + HG groups (*p* < 0.05, compared to those in the sham group). However, the higher levels of pro-inflammatory cytokines and lower level of IL-10 was noted in the ALI + HG group than in the ALI group (*p* < 0.05). The pretreatment or post-ALI treatment with the NKCC1 inhibitor reduced the levels of these pro-inflammatory cytokines and increased the levels of IL-10 in rats from both ALI and ALI + HG groups (*p* < 0.05, comparing ALI vs. ALI + pre-B, ALI + HG vs. ALI + HG + pre-B or post-B groups).

### 2.5. Effects of Hyperglycemia and NKCC1 Inhibitor Treatment on Macrophage Subsets

The distribution of AM1 and AM2 was shown in Figure 5. The sections from rats of the sham (Figure 5A) and sham + HG (Figure 5B) groups showed a rare distribution of AM1. Higher AM1 infiltration was noted in the rats of the ALI (Figure 5C) and ALI + HG (Figure 5D) groups. Decreased AM1 infiltration was noted in rats of the ALI-pre-B (Figure 5E), ALI + HG + pre-B (Figure 5F) and ALI + HG + post-B groups (Figure 5G). The AM1 counts (Figure 5H) were higher in rats with ALI and further increased in the ALI + HG group (*p* < 0.05). The NKCC1 inhibitor reduced AM1 counts in both ALI and ALI + HG groups (*p* < 0.05). The AM2 infiltration was rare in the sham (Figure 5I), sham + HG (Figure 5J), ALI (Figure 5K) and ALI + HG (Figure 5L) groups. Increased AM2 infiltration was noted in rats of the ALI-pre-B (5M), ALI + HG + pre-B (Figure 5N) and ALI + HG + post-B groups (Figure 5O). The AM2 counts (Figure 5P) were low in rats of sham, sham +HG, ALI and ALI + HG. The NKCC1 inhibitor increased AM2 counts in both ALI and ALI + HG groups (*p* < 0.05).

### 2.6. Effects of Hyperglycemia and NKCC1 Inhibitor Treatment on SGK1, NKCC1, WNK4 and SPAK

The levels of SGK1 (Figure 6A) were significantly higher in hyperglycemic than in non-hyperglycemic rats (*p* < 0.05, comparing the sham vs. sham + HG groups, the ALI vs. ALI + HG groups, and ALI + pre-B vs. ALI + HG + pre-B or post-B groups). High levels of total NKCC1 (Figure 6B) and phosphorylated NKCC1 (Figure 6C) were observed in the ALI group (*p* < 0.05, comparing the sham group), which increased further in the ALI + HG group (*p* < 0.05, comparing the ALI group). Pretreatment or post-ALI treatment of the NKCC1 inhibitor decreased the expression of NKCC1 (*p* < 0.05). The levels of WNK4 (Figure 6D) and SPAK (Figure 6E) were higher in rats of ALI and ALI + HG (*p* < 0.05, comparing the sham group) and were decreased in the rats of ALI receiving pre-treatment of post-ALI treatment of the NKCC1 inhibitor (*p* < 0.05, comparing the ALI group).

## 3. Discussion

This study states several important findings. Severe pulmonary edema, increased expression of pro-inflammatory cytokines, neutrophils, and AM1 infiltration, and the activation of the WNK4–SPAK–NKCC1 pathway were observed in rats with ALI. Rats from the ALI + HG group presented with lung injury and pulmonary edema of higher severity, along with lung inflammation, higher levels of pro-inflammatory cytokines, and upregulation of the SGK1–NKCC1 pathway. Higher levels of neutrophils and AM1 were also detected in the lung tissues. Administration of the NKCC1 inhibitor led to a significant reduction in pulmonary edema, lung inflammation, the levels of pro-inflammatory cytokines, and neutrophilic and AM1 infiltration. Higher AM2 infiltration and anti-inflammatory cytokines were also noted. These results suggest that acute hyperglycemia aggravates lung injury via the further activation of the SGK1–NKCC1 pathway. NKCC1 inhibitors concomitantly inhibit the SGK1–NKCC1 and WNK4–SPAK–NKCC1 pathways, decreased AM1, increased AM2 and reduce lung injury. This is the first study to address the mechanism underlying the aggravation of acute hyperglycemia-related lung injury by over-expression of NKCC1.

It has been demonstrated that injurious mechanical ventilation rapidly activates AMs, which subsequent play an important role in the initial pathogenesis of ALI [13]. The inflammation is related to the levels of AM1, and this process is associated with the inflammatory stage of ALI [10]. Upon exposure to lung injury insults, AM1 produces pro-inflammatory cytokines that recruit neutrophils to the lungs. Activated AM1 and neutrophils release proteases and oxidants, which further results in lung damage [10]. In the current study, we revealed that ALI presented with AM1 infiltration and increased levels of pro-inflammatory cytokines and neutrophils. The WKN4–SPAK–NKCC1 pathway is activated by the pro-inflammatory cytokines released during ALI. In rats with ALI + HG, the SGK1–NKCC1 pathway is further activated and results in higher levels of pro-inflammatory cytokines, increased neutrophil and AM1 infiltration.

The alveolar epithelium maintains the pulmonary fluid balance and facilitates optimal gas exchanges [3]. Failure to remove excess fluid from the alveoli leads to flooding of the alveolar space and impairs pulmonary gas exchange, as observed in ALI. Pulmonary NKCC1 drives water transport via the regulation of Na^+^ and Cl^−^ transported back to the alveolar epithelial cells [3]. Therefore, the activation of the WNK4–SPAK–NKCC1 pathway in lung injury hinders alveolar fluid clearance, increases alveolar fluid flooding, and causes lung injury of greater severity [4,5]. In this study, we further revealed that NKCC1 expression increased in hyperglycemia owing to the activation of the SGK1–NKCC1 pathway. The increase in NKCC1 expression further aggravated pulmonary edema and lung inflammation.

Besides its function in the regulation of alveolar fluid levels, the upregulation of NKCC1 leads to dysregulation of fluid transport, and consequently causes cell swelling and inflammation [14,15]. Cell swelling leads to apoptosis and cellular membrane rupture [16]. Resultantly, the apoptotic cells release inflammatory mediators and cytokines [16]. Activated AM1 also releases inflammatory cytokines in lung injuries [17]. Pro-inflammatory cytokines can activate the WNK4–SPAK–NKCC1 pathway, which upregulates NKCC1 expression [4]. The upregulation of NKCC1 expression results in the activation of AM1 [17] and increased expression of pro-inflammatory cytokines released from AM1. These results highlight the vicious cycle typical to lung injury.

After the administration of the NKCC1 inhibitor, the severity of lung injury reduced with concurrent changes in the phenotypes of AMs (lower AM1 and increased AM2 levels). Lin et al. suggested that NKCC1 modulation can alter the inflammatory cascade in an oxidative stress-induced lung injury [4]. Hung et al. demonstrated that NKCC1 can amplify the inflammatory response by stimulating the functions of AMs [17]. They demonstrated that bumetanide modulates NKCC1 expression, inhibits cytokine production by macrophages, and attenuates lung inflammation and tissue damage [17]. In the current study, we demonstrated that the NKCC1 inhibitor reduced the levels of AM1 and those of AM1-related pro-inflammatory cytokines. Moreover, the NKCC1 inhibitor also increased the levels of AM2 and AM2-related anti-inflammatory cytokines. Increasing AM2 can accelerate the resolution of inflammation after ALI [10]. NKCC1 inhibitors can inhibit the WNK4–SPAK–NKCC1 and SGK1–NKCC1 pathways and restore alveolar fluid clearance and inflammation cascade in lung injury.

This study is the first to reveal that hyperglycemia activates the SGK1–NKCC1 pathway to increase the expression of NKCC1 and consequently aggravates ALI. However, certain studies have been conducted on hyperglycemia and the SGK-1–NKCC1 pathway in other organs. Hills et al. showed that SGK1 expression is upregulated in the cortical collecting duct cell line of humans under hyperglycemic conditions [11]. A previous study demonstrated the increase in NKCC expression in the blood–brain barrier in hyperglycemia, which subsequently increased brain edema [18]. The upregulation of the SGK1–NKCC1 pathway in hyperglycemia consequently increased Na^+^ transporter activities in the blood–brain barrier and exacerbated cerebral edema [18]. Our study showed that the SGK1–NKCC1 pathway is also important in hyperglycemia-related aggravation of lung injury. The present study suggests that hyperglycemia augments the expression of the SGK1–NKCC1 pathway, which is one of the mechanisms of lung injury aggravation. Klug et al. suggested that the NKCC1 inhibitor was effective in treating cerebral edema [18]. Similarly, we also demonstrated that the inhalational bumetanide inhibits NKCC1 and effectively reduces pulmonary edema and inflammation in ALI. The mechanisms underlying acute hyperglycemia and effects of the NKCC1 inhibitor are shown in Figure 7.

### 3.1. Clinical Implications

Acute hyperglycemia may be associated with poorer outcomes in ICUs [7]. Traditionally, hyperglycemia can be managed to control the levels of blood glucose. However, the benefits of intensive glycemic control are subject to controversy. Berghe et al. observed that intensive insulin therapy resulted in reduced mortality in patients admitted to surgical ICUs [19]. However, in a subsequent study, this outcome was not replicated in patients who were admitted to medical ICUs [20]. A meta-analysis revealed that intensive insulin therapy did not affect the mortality in critically ill patients [21]. Therefore, controlling blood glucose levels could not alleviate the survival rates completely and the development of other effective therapeutic strategies is necessary to improve outcomes. Based on the findings from the current study, SGK1–NKCC1 is upregulated in acute hyperglycemia with lung injury. Inhibition of NKCC1 results in the improvement of pulmonary edema and lung inflammation. We believe that this treatment strategy has potential and should be investigated further.

### 3.2. Limitations of Study

Although our study clearly addressed the role of NKCC1 in ALI with acute hyperglycemia, this study has certain limitations. First, the study was conducted on rats, and thus, the results need to be confirmed in humans in future. However, our results lead the way for such clinical trials. Second, we studied the role of NKCC1 in acute hyperglycemia. However, the results may differ in chronic hyperglycemia, which suggests that the results cannot be extrapolated for patients with chronic hyperglycemia. In fact, a previous study has reported the induction of protective cellular conditioning mechanisms, such as the downregulation of glucose transporters that protect cells from glucose ingress in chronic hyperglycemia [7]. This reaction does not occur in acute hyperglycemia. Third, bumetanide is an NKCC inhibitor and loop diuretics [22]. Therefore, the diuretic effects could not be completely excluded in this study. However, rather than using bumetanide systemically, we used inhalational bumetanide with local effects in the alveolar epithelium and resultantly, the urine outputs did not differ significantly between the treatment groups. Moreover, we investigated the inflammatory response, which is not related to the diuretic effects.

## 4. Materials and Methods

### 4.1. Animal Care

This research protocol (identification code 105-IACUC-023) was approved by the Animal Review Committee of the National Defense Medical Center (NDMC, Taipei, Taiwan) on 21 June 2017. The animals used in the study were cared according to the “Guidelines for the Care and Use of Laboratory Animals”.

There were seven groups with eight animals in each group. However, ten other rats died before the completion of the experiments. Therefore, a total of 66 rats were used in this study. At the end of the experiments, exsanguination was performed by withdrawing blood from the heart under deep anesthetization.

### 4.2. ALI Model and Experiment Protocols

Male Sprague–Dawley rats (8–10 weeks, 300 ± 50 g) were used. Tracheastomy was performed after deep anesthetization was accomplished via intraperitoneal injection of sodium pentobarbital (30 mg/kg), and a PE-240 catheter (outer diameter 2.42 mm) was inserted and connected to a mechanical ventilator (modle7025, Ugo Basile, Comerio VA, Italy). The initial settings of the mechanical ventilator were: tidal volume (V_T_): 8 mL/kg, positive end-expiratory pressure (PEEP): 2 cm H_2_O, fraction of inspired oxygen (FiO_2_): 21%, and respiratory frequency (Rf): 60 cycle/min. The airway pressure was monitored during the experiment. Dynamic lung compliance (Cdyn) was calculated by dividing V_T_ by the pressure difference between peak airway pressure and PEEP [23]. A catheter (PE-50, outer diameter 0.965 mm) was inserted into the femoral artery and was connected to a pressure sensor (BSL MP45 System, BIOPAC Systems Inc., Goleta, CA, USA) to monitor the blood pressure during the experiment. Another catheter (PE-50) was inserted into the femoral vein for continuous infusion of 40% glucose water.

The ALI model was a ventilation-induced lung injury model with a V_T_ of 20 mL/kg for four hours (Figure 8) [24]. The PEEP, FiO_2_, and Rf were maintained as initial settings. Acute hyperglycemia was induced by injection with 0.5 mL 40% glucose solution followed by continuous infusion (2 mL/h) via the femoral vein. The blood glucose levels were monitored every hour (Rightest GM700S glucometer, Bionime Corporation, Taichung, Taiwan). The animals were divided into Group I, sham; Group II, sham+ HG; Group III, ALI; Group IV, ALI + HG; Group V, ALI + pretreatment of inhalational bumetanide (ALI + pre-B); Group VI, ALI+ HG + pretreatment of inhalational bumetanide (ALI + HG + pre-B), and Group VII, ALI + HG + post-ALI inhalational bumetanide (ALI + HG + post-B). Bumetanide is an NKCC inhibitor [22]. The groups V and VI were pretreated with inhalational bumetanide (0.3 mg/kg) 30 min before ALI was induced and group VII was treated 30 min after ALI was induced.

After completion of the experiment, the left lungs of each animal were used for lavage and the right lungs were used for calculating the wet/dry weight (W/D) ratio and biochemical analysis. In addition, there were two other rats in each group that were used for the examination of lung histopathology.

### 4.3. Measurement of Pulmonary Edema

The LW/BW and W/D ratios indicate pulmonary edema. At the end of the experiment, the lungs were excised and weighed to determine the LW. A portion of the right middle lobe was weighed (wet weight) and dried in an oven at 60 °C for 48 h and weighed again (dry weight). The W/D ratio was calculated based on the dry and wet weights of the lungs.

### 4.4. Determination of TP Concentration in BALF

After completion of the experiment, the right bronchus and the pulmonary artery were tied together. Then, the left lung was lavaged twice with 2.5 mL of saline. The obtained fluid was centrifuged at 200× *g* for 10 min. The TP concentration in the supernatant was determined using a commercial protein assay (Pierce, Rockford, IL, USA).

### 4.5. Histopathological Analysis of Lung Tissue

The lung tissues were harvested and fixed with 10% formaldehyde, which was infused through the trachea. We removed the lungs, which were then soaked in formaldehyde for 24 h. The lung tissues were mounted in paraffin wax using a microtome to cut sections 4–6 µm in thickness. The tissues were stained with hematoxylin and eosin (H&E).

### 4.6. Quantification of Neutrophils in Lungs

We performed H&E staining to determine the neutrophil counts in the lung tissues. In each slide, we counted the neutrophils using ten high power fields (400×).

### 4.7. ALI Score

ALI score was calculated according to (1) infiltration into or aggregation of neutrophils in the airspace or vessel walls and (2) thickness of the alveolar wall. These two observations were scored from 0 (normal) to 5 (most severe injury or greatest thickness, respectively) [17].

### 4.8. Expression of Pro-Inflammatory and Anti-Inflammatory Cytokines in the Serum

Blood was withdrawn from the femoral artery catheters and was centrifuged at 1000× *g* for 20 min to obtain the sera. The expressions of cytokines, including TNF-α, IL-1β, and CINC-1, and IL-10 in the sera were measured using enzyme-linked immunosorbent assay (ELISA) (R&D Systems Inc., Minneapolis, MN, USA).

### 4.9. Detection of AMs in Lung Tissues

Immunohistochemistry staining was conducted to detecting AM1 and AM2 [25]. The deparaffinized and rehydrated section was treated with 3% hydrogen peroxide for 15 min to eliminate endogenous peroxidase activity. Nonspecific binding sites were blocked with 3–5% bovine serum albumin (BIONOVAS, Toronto, ON, Canada) for 30 min. The section was incubated with antibodies (dilution 1:100; BioLegend, San Diego, CA, USA) raised for rabbit type-I arginase (Arg-I) of AM1 and inducible nitric oxide synthase (iNOS) of AM2 for overnight at 4 °C. The biotinylated secondary antibodies were added at a ratio of 1:250 and final signal was expressed by avidin–biotin peroxidase technique in the presence of hydrogen peroxide. The sections were evaluated under light microscope OLYMPUS IX 81 microscope (Olympus, Tokyo, Japan) and the images were analyzed with Image J software (National Institutes of Health, Bethesda, MD, USA).

Quantification of AM1 and AM2 in lungs was performed. In each slide of 200x power filed, we counted the number of AM1 and AM2 in each group.

### 4.10. Immunoblotting of SGK1, Phosphorylated, and Total NKCC1, WNK4, and SPAK

The cytoplasmic and nuclear proteins were extracted from the lung tissues using an extraction kit (BioVision, Inc., Mountain View, CA, USA). Equal quantities of lung homogenates (30 μg/lane) were fractionated using 10–12% SDS-PAGE gels and transferred to Hybond polyvinylidene fluoride membranes. The blots were incubated overnight at 4 °C with antibodies against SGK1, and total and phosphorylated NKCC1, WNK4, and SPAK. The bands were identified using enhanced chemiluminescence reagents and radiography films.

### 4.11. Data Analysis

We used SPSS version 24.0 (SPSS Inc., Chicago, IL, USA) to conduct statistical analyses. All the values were expressed in terms of mean ± standard deviation (SD). The differences between the groups were analyzed using one-way analysis of variance (ANOVA). The post-hoc analyses for the intergroup comparison were performed using Games–Howell tests. *p* < 0.05 was considered statistically significant.

## 5. Conclusions

ALI presents with severe pulmonary edema, inflammation, increased infiltration of neutrophils and AM1, and activation of the WNK4–SPAK–NKCC1 pathway. The regulation of alveolar fluid is important in ALI. High NKCC1 expression impairs the removal of alveolar fluid and leads to an increase in the levels of AM1 and AMI1-related cytokines, which subsequently aggravates lung edema and inflammation. The SGK1–NKCC1 pathway is upregulated in acute hyperglycemia, which further increases the expression of NKCC1 and the severity of ALI. Administration of an NKCC1 inhibitor can inhibit both WNK4–SPAK–NKCC1 and SGK1–NKCC1 pathways and aid in the attenuation of lung injury. Clinically, NKCC1 may serve as a target in pharmacological interventions for ALI.

## Figures and Tables

**Figure 1 ijms-21-04803-f001:**
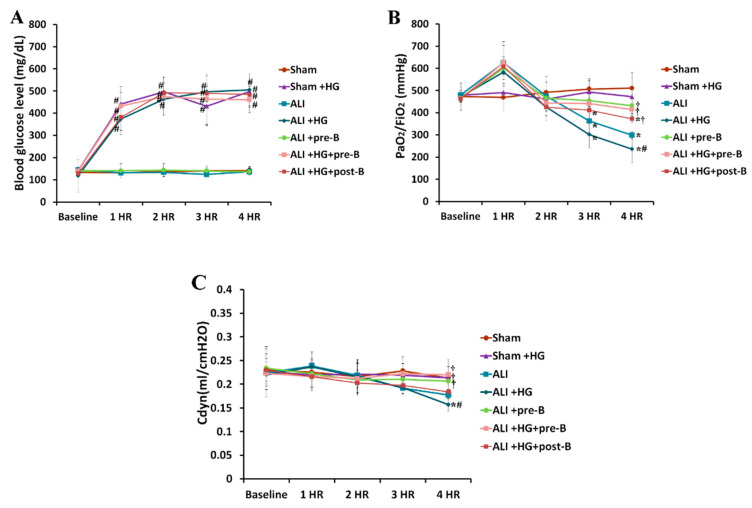
Blood glucose levels, oxygenation, and dynamic compliance of lungs. The blood glucose levels (**A**) in the hyperglycemic rats were 400–500 mg/dL, which were significantly higher than that in non-hyperglycemic rats (*p* < 0.05). PaO_2_/FiO_2_ (**B**) was significantly reduced in rats with ALI or ALI + HG (*p* < 0.05). PaO_2_/FiO_2_ was lower in rats with ALI + HG than in rats with ALI (*p* < 0.05). The rats that received the NKCC1 inhibitor had higher PaO_2_/FiO_2_ than those without NKCC1 inhibitor (*p* < 0.05). The Cdyn (**C**) was significantly reduced in rats with ALI or ALI + HG than in those from the sham group (*p* < 0.05). The Cdyn was lower in rats with ALI + HG than in those with ALI (*p* < 0.05). The rats with the NKCC1 inhibitor had higher Cdyn than those without the NKCC1 inhibitor (*p* < 0.05). * Significant difference, as compared to the sham group (*p* < 0.05), ^#^ Significant difference, comparing the groups with HG to those without HG (I vs. II, III vs. IV, V vs. VI, and V vs. VII, *p* < 0.05), ^†^ Significant difference, comparing the ALI groups with B to those without B (III vs. V, IV vs. VI, and IV vs. VII, *p* < 0.05) Abbreviation: PaO_2_ = partial pressure of oxygen in arterial blood, FiO_2_ = fraction of inspired oxygen, Cdyn = dynamic compliance of lungs, ALI = acute lung injury, HG = hyperglycemia, pre-B = pretreatment with bumetanide, post-B = post-treatment with bumetanide.

**Figure 2 ijms-21-04803-f002:**
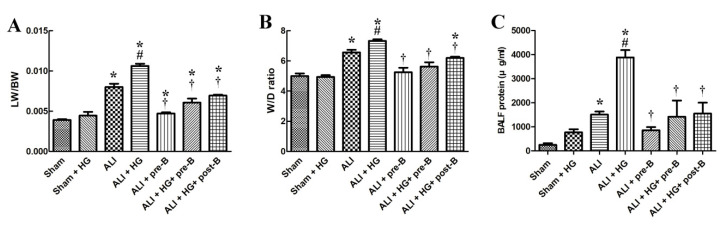
Effects of hyperglycemia and NKCC1 inhibitor treatment on pulmonary edema and alveolar protein leakage. The LW/BW (**A**) and W/D ratios (**B**) were significantly increased in rats with ALI and further increased in rats with ALI + HG (*p* < 0.05). The NKCC1 inhibitor reduced pulmonary edema in both ALI and ALI + HG groups (*p* < 0.05). The TP concentration in BALF (**C**) was prominent in rats with ALI and further increased in rats with ALI + HG (*p* < 0.05). The NKCC1 inhibitor reduced alveolar protein leakage in both ALI and ALI + HG rats (*p* < 0.05). * Significant difference, as compared to the sham group (*p* < 0.05), ^#^ Significant difference, comparing the groups with HG to those without HG (I vs. II, III vs. IV, V vs. VI, and V vs. VII, *p* < 0.05), ^†^ Significant difference, comparing the ALI groups with B to those without B (III vs. V, IV vs. VI, and IV vs. VII, *p* < 0.05). Abbreviation: LW/BW = lung weight/body weight, W/D = lung wet/dry weight ratio. ALI = acute lung injury, HG = hyperglycemia, pre-B = pretreatment with bumetanide, post-B = post-treatment with bumetanide, BALF = bronchoalveolar lavaged fluid, TP = total protein.

**Figure 3 ijms-21-04803-f003:**
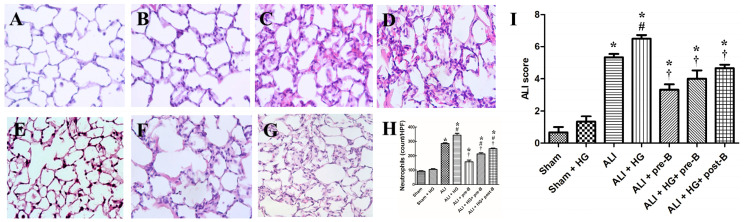
Effects of hyperglycemia and NKCC1 inhibitor treatment on the histopathological features in ALI and neutrophil sequestration. The sham (**A**) and sham + HG (**B**) groups exhibited normal histology. Sections from rats of the ALI (**C**) and ALI + HG (**D**) groups revealed severe lung injury with prominent alveolar exudation, alveolar septum thickening, and neutrophil sequestration. Pretreatment or post-ALI treatment with the NKCC1 inhibitor reduced lung injury, which was observable in (**E**) ALI + pre-B, (**F**) ALI + HG + pre-B, and (**G**) ALI + HG + post-B groups. The neutrophil counts (**H**) were high in rats with ALI and further increased in the ALI + HG group (*p* < 0.05). The NKCC1 inhibitor reduced neutrophil counts in both ALI and ALI + HG groups (*p* < 0.05). The ALI scores (**I**) were high in rats with ALI and further increased in the ALI + HG group (*p* < 0.05). The NKCC1 inhibitor reduced ALI scores in both ALI and ALI + HG groups (*p* < 0.05). * Significant difference, as compared to the sham group (*p* < 0.05), ^#^ Significant difference, comparing the groups with HG to those without HG (I vs. II, III vs. IV, V vs. VI, and V vs. VII, *p* < 0.05), ^†^ Significant difference, comparing the ALI groups with B to those without B (III vs. V, IV vs. VI, and IV vs. VII, *p* < 0.05). Abbreviation: ALI = acute lung injury, HG = hyperglycemia, pre-B = pretreatment with bumetanide, post-B = post-treatment with bumetanide.

**Figure 4 ijms-21-04803-f004:**
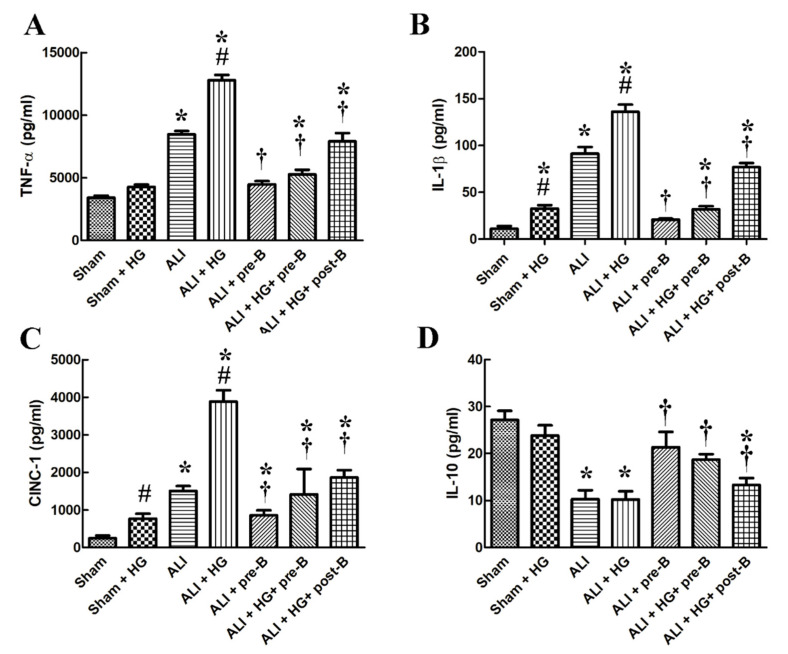
Effects of hyperglycemia and NKCC1 inhibitor treatment on the levels of cytokines. Higher levels of serum TNF-α (**A**), IL-1β (**B**), CINC-1, (**C**) and lower level of IL-10 (**D**) were observed in rats from the ALI and ALI + HG groups (*p* < 0.05). However, the higher levels of pro-inflammatory cytokines and lower level of anti-inflammatory cytokine was noted in the ALI + HG group than in the ALI group (*p* < 0.05). The NKCC1 inhibitor reduced the levels of these pro-inflammatory cytokines and increased the levels of anti-inflammatory cytokine in rats from both ALI and ALI + HG groups (*p* < 0.05). * Significant difference, as compared to the sham group (*p* < 0.05), ^#^ Significant difference, comparing the groups with HG to those without HG (I vs. II, III vs. IV, V vs. VI, and V vs. VII, *p* < 0.05), ^†^ Significant difference, comparing the ALI groups with B to those without B (III vs. V, IV vs. VI, and IV vs. VII, *p* < 0.05). Abbreviation: TNF-α = tumor necrosis factor alpha, IL-1β = interleukin 1 beta, CINC-1 = cytokine-induced neutrophil chemoattractant 1, IL-10 = interleukin 10, ALI = acute lung injury, HG = hyperglycemia, pre-B = pretreatment with bumetanide, post-B = post-treatment with bumetanide.

**Figure 5 ijms-21-04803-f005:**
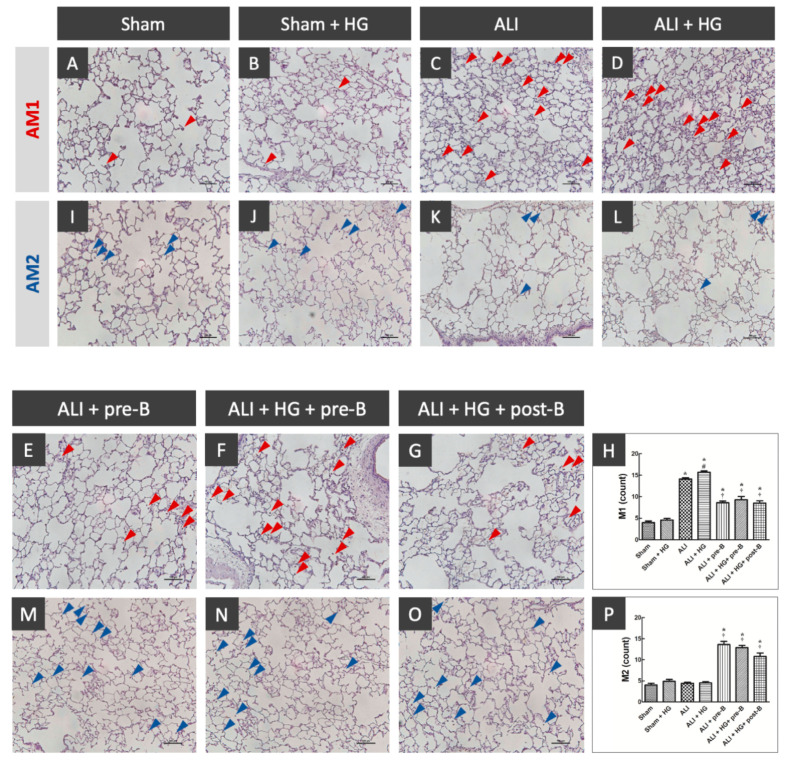
Effects of hyperglycemia and NKCC1 inhibitor treatment on macrophages. The sections collected from rats of the sham (**A**) and sham + HG (**B**) groups showed rare distribution of AM1 (red arrow). Higher AM1 infiltration was noted in the rats of the ALI (**C**) and ALI + HG (**D**) groups. Decreased AM1 infiltration was noted in rats of the ALI-pre-B (**E**), ALI + HG + pre-B (**F**) and ALI + HG + post-B groups (**G**). The AM1 counts (**H**) were higher in rats with ALI and further increased in the ALI + HG group (*p* < 0.05). The NKCC1 inhibitor reduced AM1 counts in both ALI and ALI + HG groups (*p* < 0.05). The AM2 infiltration (blue arrow) was rare in the sham (**I**), sham + HG (**J**), ALI (**K**) and ALI + HG (**L**) groups. Increased AM2 infiltration was noted in rats of the ALI + pre-B (**M**), ALI + HG + pre-B (**N**) and ALI + HG + post-B groups (**O**). The AM2 counts (**P**) were low in rats of sham, sham +HG, ALI and ALI + HG. The NKCC1 inhibitor increased AM2 counts in both ALI and ALI + HG groups (*p* < 0.05). * Significant difference, as compared to the sham group (*p* < 0.05), ^#^ Significant difference, comparing the groups with HG to those without HG (I vs. II, III vs. IV, V vs. VI, and V vs. VII, *p* < 0.05), ^†^ Significant difference, comparing the ALI groups with B to those without B (III vs. V, IV vs. VI, and IV vs. VII, *p* < 0.05). Abbreviation: AM1 = alveolar macrophage 1, AM2 = alveolar macrophage 2, ALI = acute lung injury, HG = hyperglycemia, pre-B = pretreatment with bumetanide, post-B = post-treatment with bumetanide.

**Figure 6 ijms-21-04803-f006:**
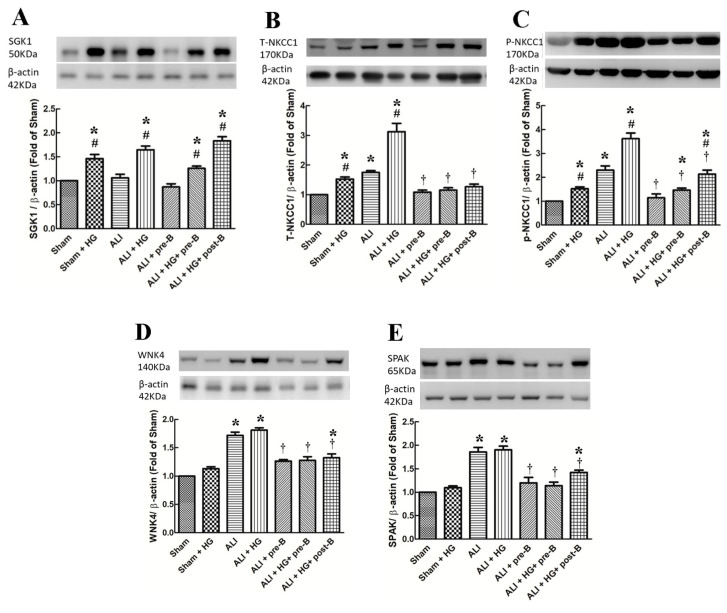
Effects of hyperglycemia and NKCC1 inhibitor treatment on the levels of SGK1, NKCC1, WNK4 and SPAK. The levels of SGK1 (**A**) were significantly higher in hyperglycemic rats than in non-hyperglycemic rats (*p* < 0.05). High levels of T-NKCC1 (**B**) and p-NKCC1 (**C**) were observed in the ALI group, which increased further in the ALI + HG group (*p* < 0.05). The NKCC1 expression was lower in the rats of ALI + pre-B, ALI + HG + pre-B and ALI + HG + post B (*p* < 0.05). The levels of WNK4 (**D**), and SPAK (**E**) were higher in rats of ALI and ALI + HG (*p* < 0.05) and decreased in the rats of ALI + pre-B, ALI + HG + pre-B and ALI + HG + post B (*p* < 0.05). * Significant difference, as compared to the sham group (*p* < 0.05), ^#^ Significant difference, comparing the groups with HG to those without HG (I vs. II, III vs. IV, V vs. VI, and V vs. VII, *p* < 0.05), ^†^ Significant difference, comparing the ALI groups with B to those without B (III vs. V, IV vs. VI, and IV vs. VII, *p* < 0.05). Abbreviation: ALI = acute lung injury, SGK1 = serum-glucocorticoid kinase 1, T-NKCC1 = total sodium-potassium-chloride co-transporter one, p-NKCC1 = phosphorylated sodium-potassium-chloride co-transporter one, WNK4 = with-no-lysine kinases 4, SPAK = STE20/SPS1-related proline/alanine-rich kinase, HG = hyperglycemia, pre-B = pretreatment with bumetanide, post-B = post-treatment with bumetanide.

**Figure 7 ijms-21-04803-f007:**
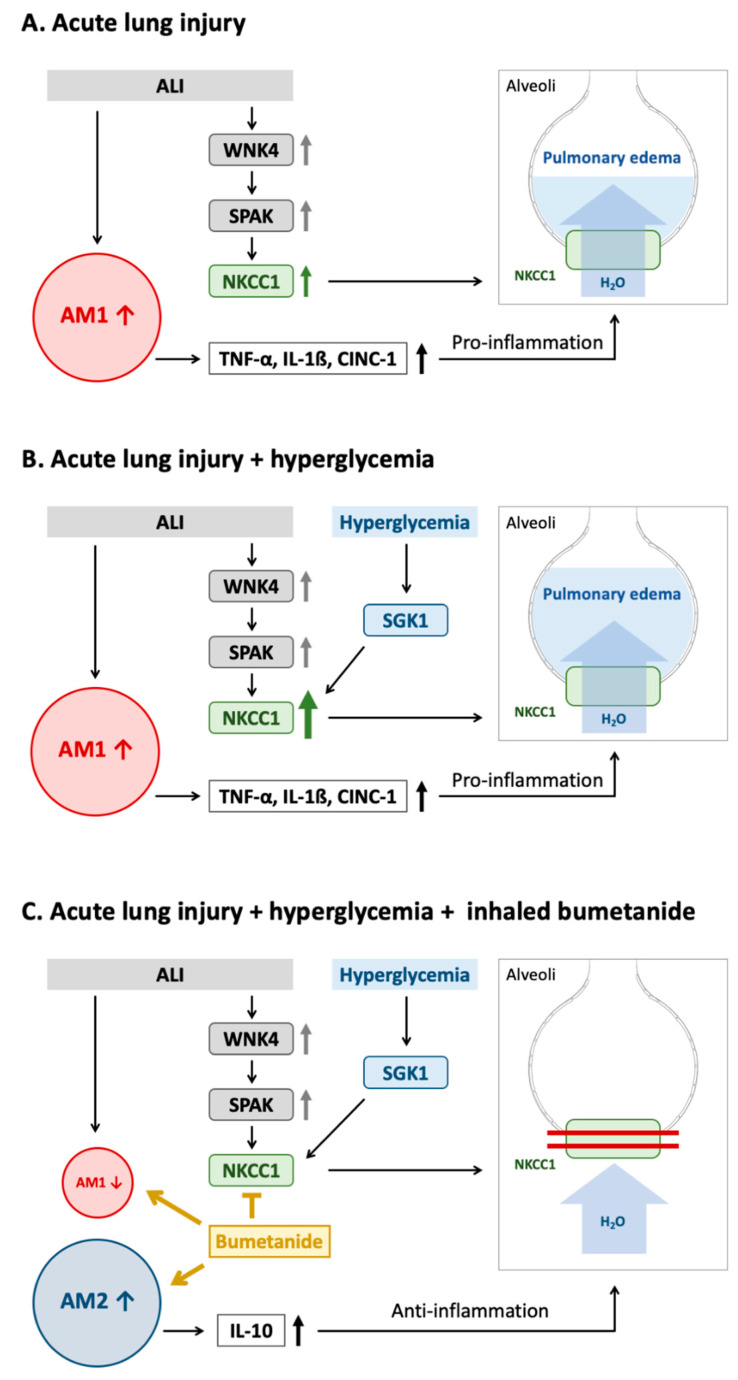
Mechanism underlying the effect of acute hyperglycemia on SGK1–NKCC1 pathway in acute lung injury. (**A**) The WNK4–SPAK–NKCC1 pathway is activated in ALI. Higher expression of NKCC1 results in the activation of AM1, increased expression of pro-inflammatory cytokines and impaired alveolar fluid clearance, which result in pulmonary edema and inflammation. (**B**) Acute hyperglycemia further activates the SGK1–NKCC1 pathway, increases expression of NKCC1 and AM1 activation that consequently results in more severe pulmonary edema, and lung inflammation. (**C**) The NKCC1 inhibitor can inhibit WNK4–SPAK–NKCC1 and SGK1–NKCC1 pathway that leads to decreased AM1 and pro-inflammatory cytokines, increased AM2 and anti-inflammatory cytokines, reduction in water influx. These result in decreasing pulmonary edema, and inflammation. Abbreviation: ALI = acute lung injury, AM1 = alveolar macrophage 1, AM2 = alveolar macrophage 2, WNK4 = with-no-lysine kinases 4, SPAK = STE20/SPS1-related proline/alanine-rich kinase, SGK1 = Serum-glucocorticoid kinase 1, NKCC1 = sodium-potassium-chloride co-transporter one, TNF-α = tumor necrosis factor alpha, IL-1β = interleukin 1 beta, CINC-1 = cytokine-induced neutrophil chemoattractant 1, IL-10 = interleukin 10.

**Figure 8 ijms-21-04803-f008:**
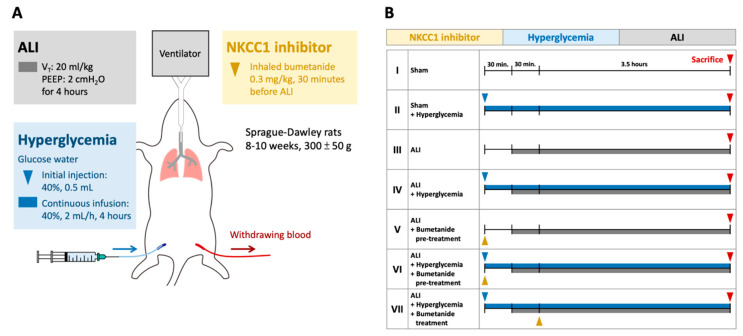
Protocol for the experiment. (**A**) The ALI model was induced by a V_T_ of 20 mL/kg for four hours. Acute HG was induced by injection with 0.5 mL 40% glucose solution followed by continuous infusion (2 mL/h). Inhalational bumetanide was administered via tracheostomy. (**B**) The animals were divided into Group I, sham; Group II, sham+ HG; Group III, ALI; Group IV, ALI + HG; Group V, ALI + pretreatment of inhalational bumetanide (ALI + pre-B); Group VI, ALI+ HG + pretreatment of inhalational bumetanide (ALI + HG + pre-B), and Group VII, ALI + HG + post-ALI inhalational bumetanide (ALI + HG + post-B). Abbreviation: ALI = acute lung injury, V_T_ = tidal volume, NKCC1 = sodium-potassium-chloride co-transporter one, HG = hyperglycemia, pre-B = pretreatment with bumetanide, post-B = post-treatment with bumetanide.

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
