# Peer review of "Acute Hyperglycemia Aggravates Lung Injury via Activation of the SGK1–NKCC1 Pathway"

_ijms, 2020, doi:10.3390/ijms21134803_

Round 1

Reviewer 1 Report

All my concerns have been satisfactory solved by the authors. The manusdript has been improved during the review proccess and I have no further comments.

----------------------------------------------------------------------------------------

Reviewer's comments and authors' responses on the original manuscript:

Comments and Suggestions for Authors

The authors of this study determined that acute hyperglycemia increases lung damage in a rat model of acute lung injury (ALI). Lung injury was determined by morphological, anatomical criteria and by the presence of total proteins in bronchoalveolar lavaged fluid and the induction of cytokines. All these ALI manifestations are decreased by inhalation of the NKCC1 (sodium-potasium-chloride-co-transporter 1) inhibitor bumetanide. The authors later showed that ALI and acute hypergycemia induced the expression of NKCC1 and that this increase is lower in the presence of bumetanide. Hyperglycemia also increases expression of serum-glucocorticoid kinase 1 (SGK1) that has been described to induce NKCC1 expression. The authors conclude that hyperglycemia inceases ALI via activation of the SGK1/NKCC1 pathway and that this process is inhibited by bumetanide.

The data presented in this article are sound and support the conclusions of the authors. The study is very interesting in the field of ALI and might have some therapeutic implications.

Major point:

The authors conclude in the article that hyperglycemia activates lung injure via activation of the SGK1/NKCC1 pathway. However, the data presented in Figure 6 do not completely support this conclusion. According to Fig6, SGK1 expression almost exclusively is activated by hyperglycemia. In contrast, NKCC1 expression and phophorylation are concomitantly activated by hyperglycemia and ALI and are decreased by bumetanide inhalation. These data indicate that NKCC1 expression might depend of other mechanisms in addition to SGK1. It is also interesting that NKCC1 expression and phosphorylation is inhibited by the NKCC1 inhibitor what might indicate the presence of feed-forward regulatory mechanisms. These possibilities should be considered by the authors.

Answer:

  1. We thank you for your valuable comments. We agreed that the previous data cannot fully explain the changes in NKCC1. During ALI, the WNK4-SPAK-NKCC1 pathway is activated. However, hyperglycemia further activates the SGK1-NKCC1 pathway and exacerbates lung injury. We addressed this issue in the revised manuscript.
  2. In our data, we showed that bumetanide decrease total and phosphorylated NKCC1 that means bumetanide inhibit expression and phosphorylation of NKCC1. This result is the similar to one previous study of Hung et al. We agreed your comments that this result indicates the presence of feed-forward regulatory mechanisms of bumetanide.

(Reference: Biochem Pharmacol . 2018 Oct;156:60-67. Bumetanide Attenuates Acute Lung Injury by Suppressing Macrophage Activation. Chin-Mao Hung, Chung-Kan Peng, Chin-Pyng Wu, Kun-Lun Huang)

Minor points:

  1. Section 2.4, first line. “and were” might be “were”

Answer: we corrected this mistake.

  1. Figure 5 legend. “TNFa(A)” should be TNFa (B)

Answer: we corrected these mistakes.

Reviewer 2 Report

I agree with the revised version of the manuscript, and have no further concern.

-------------------------------------------------------------------------------

Comments and Authors' Responses on the original manuscript:

Comments and Suggestions for Authors

The manuscript of Wu et al. (entitled Acute hyperglycemia aggravates lung injury via activation of the SGK1-NKCC1 pathway) shows the worsening of acute lung injury (ALI) by hyperglycemia via the SGK1-NKCC1 pathway in an experimentally induced ALI-model in rats.

The subject of this work would be of interest and the present manuscript provides solid data for a better understanding of the pathophysiology of ALI in acute hyperglycemia.

Answer: Thank you for your comments. This is the first study that addressed the influence of acute hyperglycemia in NKCC1 in ALI.

The experimental procedures and the statistics are appropriate.

Answer: Thank you for your comments. We did our best to design the study and perform the experiment.

The results are clearly presented, the conclusion is correct, but there are some points for criticism that must be addressed in a new submission.

Answer: We did our best to response the comments. We performed more experiments and addressed a lot of new findings in the revised manuscript.

Major points:

The effect of hyperglycemia on the expression of SGK1, total NKCC1 and phosphorylated NKCC1 is convincingly demonstrated in Fig.6. In the same figure, however, there is no significant change of the SGK1 expression in ALI as compared to sham (panel A, first and third columns), although there are significant changes in respect of the NKCC1 expression. How do the authors interprete these findings regarding the pathophysiology of ALI? In addition, at least one block of representative images of immunoblots must be shown, and not only the densitometric analyses of them (Fig.6).

Answer:

  1. We thank you for your valuable comments. We agree that the previous data cannot fully explain the changes in NKCC1 in lung injury with or without hyperglycemia. During ALI, the WNK4-SPAK-NKCC1 pathway is activated. However, hyperglycemia further activates the SGK1-NKCC1 pathway and exacerbates lung injury. We addressed this issue in the revised manuscript. We provided more information about mechanism of hyperglycemia related aggravation of lung injury in Fig. 8.
  2. We further provided the images of immunoblots as your valuable suggestions.

The figure legends of Figs. 3-4-5-6 are extremely long, containing lots of results, which are repeated in the text-body again. The authors must eliminate useless repetitions.

Answer: We shorten the descriptions in the figure legends as your suggestions.

Minor point:

The title of the Fig.6 indicates data of pro-inflammatory cytokines (likely a typing error), but there are no cytokine-results in this figure at all.

Answer: We are sorry for the careless mistakes. We corrected this mistake. Besides, we provided the data of anti-inflammatory cytokines.

This manuscript is a resubmission of an earlier submission. The following is a list of the peer review reports and author responses from that submission.

Round 1

Reviewer 1 Report

The authors of this study determined that acute hyperglycemia increases lung damage in a rat model of acute lung injury (ALI). Lung injury was determined by morphological, anatomical criteria and by the presence of total proteins in bronchoalveolar lavaged fluid and the induction of cytokines. All these ALI manifestations are decreased by inhalation of the NKCC1 (sodium-potasium-chloride-co-transporter 1) inhibitor bumetanide. The authors later showed that ALI and acute hypergycemia induced the expression of NKCC1 and that this increase is lower in the presence of bumetanide. Hyperglycemia alo increases expression of serum-glucocorticoid kinase 1 (SGK1) that has been described to induce NKCC1 expression. The authors conclude that hyperglycemia inceases ALI via activation of the SGK1/NKCC1 pathway and that this process is inhibited by bumetanide.

The data presented in this article are sound and support the conclusions of the authors. The study is very interesting in the field of ALI and might have some therapeutic implications.

Major point:

The authors conclude in the article that hyperglycemia activates lung injure via activation of the SGK1/NKCC1 pathway. However, the data presented in Figure 6 do not completely support this conclusion. According to Fig6, SGK1 expression almost exclusively is activated by hyperglycemia. In contrast, NKCC1 expression and phophorylation are concomitantly activated by hyperglycemia and ALI and are decreased by bumetanide inhalation. These data indicate that NKCC1 expression might depend of other mechanisms in addition to SGK1. It is also interesting that NKCC1 expression and phosphorylation is inhibited by the NKCC1 inhibitor what might indicate the presence of feed-forward regulatory mechanisms. These possibilities should be considered by the authors.

Minor points:

  1. Section 2.4, first line. “and were” might be “were”
  2. Figure 5 legend. “TNFa(A)” should be TNFa (B)

Author Response

Comments and Suggestions for Authors

The authors of this study determined that acute hyperglycemia increases lung damage in a rat model of acute lung injury (ALI). Lung injury was determined by morphological, anatomical criteria and by the presence of total proteins in bronchoalveolar lavaged fluid and the induction of cytokines. All these ALI manifestations are decreased by inhalation of the NKCC1 (sodium-potasium-chloride-co-transporter 1) inhibitor bumetanide. The authors later showed that ALI and acute hypergycemia induced the expression of NKCC1 and that this increase is lower in the presence of bumetanide. Hyperglycemia also increases expression of serum-glucocorticoid kinase 1 (SGK1) that has been described to induce NKCC1 expression. The authors conclude that hyperglycemia inceases ALI via activation of the SGK1/NKCC1 pathway and that this process is inhibited by bumetanide.

The data presented in this article are sound and support the conclusions of the authors. The study is very interesting in the field of ALI and might have some therapeutic implications.

Major point:

The authors conclude in the article that hyperglycemia activates lung injure via activation of the SGK1/NKCC1 pathway. However, the data presented in Figure 6 do not completely support this conclusion. According to Fig6, SGK1 expression almost exclusively is activated by hyperglycemia. In contrast, NKCC1 expression and phophorylation are concomitantly activated by hyperglycemia and ALI and are decreased by bumetanide inhalation. These data indicate that NKCC1 expression might depend of other mechanisms in addition to SGK1. It is also interesting that NKCC1 expression and phosphorylation is inhibited by the NKCC1 inhibitor what might indicate the presence of feed-forward regulatory mechanisms. These possibilities should be considered by the authors.

Answer:

  1. We thank you for your valuable comments. We agreed that the previous data cannot fully explain the changes in NKCC1. During ALI, the WNK4-SPAK-NKCC1 pathway is activated. However, hyperglycemia further activates the SGK1-NKCC1 pathway and exacerbates lung injury. We addressed this issue in the revised manuscript.
  2. In our data, we showed that bumetanide decrease total and phosphorylated NKCC1 that means bumetanide inhibit expression and phosphorylation of NKCC1. This result is the similar to one previous study of Hung et al. We agreed your comments that this result indicates the presence of feed-forward regulatory mechanisms of bumetanide.

(Reference: Biochem Pharmacol . 2018 Oct;156:60-67. Bumetanide Attenuates Acute Lung Injury by Suppressing Macrophage Activation. Chin-Mao Hung, Chung-Kan Peng, Chin-Pyng Wu, Kun-Lun Huang)

Minor points:

  1. Section 2.4, first line. “and were” might be “were”

Answer: we corrected this mistake.

  1. Figure 5 legend. “TNFa(A)” should be TNFa (B)

Answer: we corrected these mistakes.

Reviewer 2 Report

Authors demonstrated that acute hyperglycemia aggravated the ventilator-induced acute lung injury in rats, and that pretreatment with inhaled NKCC1 inhibitor bumetanide significantly decreased pulmonary edema and inflammation.

The manuscript is clearly and concisely written, however, I have some objections or comments to the authors.

Abstract, line 8: Explain shortly in abstract why you decided to use NKCC1 inhibitor – in relation to hyperglycemia and ALI.  It is nicely explained by the sentences in the Introduction, par. 4: „... However, hyperglycemia is known to...  via activation of the SGK1-NKCC1 pathway.“ You can use shorter version of these sentences.

Abstract, line 12: I suggest to add that bumetanide is an inhaled NKCC1 inhibitor, when it is firstly mentioned in abstract.

Abstract, line 16: I suggest to change: „...The administration of an inhaled NKCC1 inhibitor...“ by „The pretreatment with an inhaled NKCC1 inhibitor“.

In whole text, please, use „glucose“ instead of „sugar“.

Results, section 2.1: I would like to suggest to divide the first paragraph of this section (2.1). Part named “ALI model“ (lines 1-9) should be moved to Methods (as section 4.2). In this paragraph, it is necessary to add abbreviations of individual groups which are later used in Figures. I also suggest to move Figure 1 do Methods.

Part „Sugar levels“ should be renamed to „Blood glucose levels“ and will begin from line 9: ... The blood sugar levels...“.

Legend to Figure 2 should be rephrased to „Blood glucose levels“. Short description is needed as well as a statistical significance of the results in this Figure. Statistical significance of these results should be added also into the Results part.

Limitations of the study: It should be mentioned that bumetanide was administered as a pretreatment 30 min before ALI what might reduce the clinical value of these results. Is it possible to expect positive results of bumetamide also when administered as a treatment , e.g., 30 min or 1 h after ALI induction?

Materials and Methods: What was a gender of animals? What were the numbers of animals in the groups?  Are you sure that diameter of the tracheal cannula is 2.5 cm, and not  2.5 mm?

How was the exsanguination performed? By blood taking from arterial catheter? It should be mentioned.

How was the lung processed in the animals? Half of the lung was lavaged (after ligature of the main bronchus?) and half of the lung was used for histology and biochemical analyses in each animal? Or some animals from the groups were lavaged, some were used for histological investigation... Which portions of lungs were used for these analyses?

How the serum was prepared?

English should be checked.

Author Response

Comments and Suggestions for Authors

Authors demonstrated that acute hyperglycemia aggravated the ventilator-induced acute lung injury in rats, and that pretreatment with inhaled NKCC1 inhibitor bumetanide significantly decreased pulmonary edema and inflammation.

The manuscript is clearly and concisely written, however, I have some objections or comments to the authors.

Abstract, line 8: Explain shortly in abstract why you decided to use NKCC1 inhibitor – in relation to hyperglycemia and ALI. It is nicely explained by the sentences in the Introduction, par. 4: „... However, hyperglycemia is known to...  via activation of the SGK1-NKCC1 pathway.“ You can use shorter version of these sentences.

Answer: We changed the sentence according to your suggestion. We wrote a lot of Abstract section of the revised manuscript.

Abstract, line 12: I suggest to add that bumetanide is an inhaled NKCC1 inhibitor, when it is firstly mentioned in abstract.

Answer: We added the description that bumetanide is a NKCC1 inhibitor.

Abstract, line 16: I suggest to change: „...The administration of an inhaled NKCC1 inhibitor...“ by „The pretreatment with an inhaled NKCC1 inhibitor“.

In whole text, please, use „glucose“ instead of „sugar“.

Answer:

  1. We changed the sentence as your suggestions. Besides of pretreatment, we added post-ALI treatment of NKCC1 inhibitor.
  2. We changed “sugar” to “glucose” according to your comments.

Results, section 2.1: I would like to suggest to divide the first paragraph of this section (2.1). Part named “ALI model “ (lines 1-9) should be moved to Methods (as section 4.2). In this paragraph, it is necessary to add abbreviations of individual groups which are later used in Figures. I also suggest to move Figure 1 do Methods.

Answer:

  1. We moved some of the first paragraph of section 2.1 to Methods section. We only showed the blood glucose levels in section 2.1. We provided other physiological data such as PaO2/FiO2 ratios and dynamic compliance in this section.
  2. We added the abbreviations of individual groups which are later used in Figures.
  3. We moved the Figure 1 to Methods.

Part „Sugar levels“ should be renamed to „Blood glucose levels“ and will begin from line 9: ... The blood sugar levels...“.

Answer: We corrected this paragraph as your suggestions.

Legend to Figure 2 should be rephrased to „Blood glucose levels“. Short description is needed as well as a statistical significance of the results in this Figure. Statistical significance of these results should be added also into the Results part.

Answer: We rephrased this figure and added other physiological data in Figure 2. We also provided statistical significance of these results in the Results section and figure legend.

Limitations of the study: It should be mentioned that bumetanide was administered as a pretreatment 30 min before ALI what might reduce the clinical value of these results. Is it possible to expect positive results of bumetamide also when administered as a treatment, e.g., 30 min or 1 h after ALI induction?

Answer: In this study, we focused on the mechanism of the influence of hyperglycemia in SGK1-NKCC1 pathway in ALI. After pretreatment of inhaled NKCC1 inhibitor, the rats were in low levels of NKCC1 and we could study the influence of hyperglycemia in animals under conditions of low levels of NKCC1. However, we still performed post-ALI treatment groups (30 minutes after lung injury) as your suggestions. According to this group, we can understand the therapeutic effects of inhaled bumetanide in lung injury.

Materials and Methods: What was a gender of animals? What were the numbers of animals in the groups? Are you sure that diameter of the tracheal cannula is 2.5 cm, and not 2.5 mm?

Answer:

  1. Male Sprague-Dawley rats (8-10 weeks, 300±50 g) were used in this study. We provided this information in revised manuscript.
  2. Tracheastomy was performed and a PE-240 catheter (outer diameter 2.42 mm) was inserted. We are sorry for the careless mistake. We corrected this mistake.

How was the exsanguination performed? By blood taking from arterial catheter? It should be mentioned.

Answer: At the end of the experiments, the exsanguination was performed by withdraw of blood from the heart under deep anesthesia. We mentioned this in the revised manuscript.

How was the lung processed in the animals? Half of the lung was lavaged (after ligature of the main bronchus?) and half of the lung was used for histology and biochemical analyses in each animal? Or some animals from the groups were lavaged, some were used for histological investigation... Which portions of lungs were used for these analyses?

How the serum was prepared?

Answer:

  1. After completion of the experiment, the right bronchus and right pulmonary artery were tied together. Then, the left lung was used for lavage. The right middle lobes were used for measurement of W/D ratio and the right lower lobes were used for immunoblotting. In addition, there were two other rats in each group for lung histopathology.
  2. The blood was withdrawn from the catheter in the femoral artery and was centrifuged at 1000 × g for 20 minutes to obtain serum.

English should be checked.

Answer: We checked and corrected the English. The revised manuscript was sent for English editing.

Reviewer 3 Report

The authors generate a spurious impression in the introduction that the accumulation of neutrophils is the major triggering factor for ALI, although many experimental results related to the function of alveolar macrophages. The authors should analyze this point in the discussion.

The authors present several data of cytokine secretion of macrophages in ALI, but “basic” parameters of these cells are missing, e.g. alveolar macrophage count, typing of activated macrophages M1, M2, etc.

the experimental protocols should be more specific, ventilatory setting, etc.

What criteria did you establish as o model (PaO2, Cdyn?) it should be mentioned in the study.

How many animals were used in this study?

Did you evaluate also the parameters of acid-base homeostasis?

Author Response

Comments and Suggestions for Authors

The authors generate a spurious impression in the introduction that the accumulation of neutrophils is the major triggering factor for ALI, although many experimental results related to the function of alveolar macrophages. The authors should analyze this point in the discussion.

Answer: We thank you for your valuable comments. We added a description of the importance of alveolar macrophages in acute lung injury in the "Introduction" section. In the "Discussion" section, we also discussed the effect of NKCC1 on macrophages.

The authors present several data of cytokine secretion of macrophages in ALI, but “basic” parameters of these cells are missing, e.g. alveolar macrophage count, typing of activated macrophages M1, M2, etc.

Answer: We thank you for your valuable comments. We performed immunohistochemistry staining to detecting AM1 and AM2 in lung tissues. We also further measured the anti-inflammatory cytokines (IL-10) of AM2. The numbers of AM1 was rare in the rats of sham or sham with hyperglycemia. However, the AM1 was significantly increased in the rats of ALI and ALI with hyperglycemia. Pretreatment of post-ALI treatment of bumetanide significantly decreased the number of AM1. The AM2 was rare in the sham, sham + HG and further decreased in the ALI and ALI + HG groups. Increased AM2 infiltration was noted in rats receiving pre-treatment or post-ALI treatment of bumetanide.

experimental protocols should be more specific, ventilatory setting, etc.

Answer: The initial settings of mechanical ventilator were tidal volume (VT) of 8 ml/kg, positive end-expiratory pressure (PEEP) of 2 cm H2O, the fraction of inspired oxygen (FiO2) of 21% and the respiratory frequency (Rf) of 60 cycle/min. The ALI model was a ventilation-induced lung injury with a tidal volume (VT) of 20 ml/kg for four hours. The PEEP, FiO2 and Rf were maintained as initial settings. We provided the information in the revised manuscript.

What criteria did you establish as model (PaO2, Cdyn?) it should be mentioned in the study?

Answer: We provided the data of PaO2/FiO2 ratios and Cdyn in Figure 1. After inducing lung injury, PaO2 / FiO2 ratio is less than 300 mmHg that meets the clinical criteria for acute lung injury. In addition, the pathological features are also compatible with lung injury.

Reference: J Postgrad Med. Oct-Dec 2011;57(4):286-90. doi: 10.4103/0022-3859.90077. Clinical Characteristics and Outcomes of Patients With Acute Lung Injury and ARDS

How many animals were used in this study?

Answer: There are seven groups with 8 animals in each group. However, 10 rats died before the experiment was completed. A total of 66 rats were used in this study. We added this description in the revised manuscript.

Did you evaluate also the parameters of acid-base homeostasis?

Answer: We measured arterial blood gas during the experiments. However, the ventilator settings of the lung injury were high tidal volume (VT, 20 ml/kg) and high respiratory frequency (Rf, 60 cycle/min). These setting of ventilator were commonly used in ventilation induced lung injury (see reference). However, these settings lead the rats in status of hyperventilation and CO2 wash out. Our data of arterial blood gas showed respiratory alkalosis with low PaCO2 (about 15 mmHg) and low HCO3- (about 10-14 meq). Therefore, the data of arterial blood gas could not present the status of lung injury. Therefore, we only showed the data of PaO2/FiO2 ratio but not PaCO2.

Reference:

  1. BMC Anesthesiol. 2018 Aug 18;18(1):116. doi: 10.1186/s12871-018-0576-7. Deferoxamine Preconditioning Ameliorates Mechanical Ventilation-Induced Lung Injury in Rat Model via ROS in Alveolar Macrophages: A Randomized Controlled Study (VT= 40 ml/kg, Rf= 40-60 breaths/min)
  2. Acta Anaesthesiol Taiwan. 2008 Dec;46(4):151-9. doi: 10.1016/S1875-4597(09)60002-3. Effects of Dexmedetomidine on Regulating Pulmonary Inflammation in a Rat Model of Ventilator-Induced Lung Injury (VT= 20 ml/kg, Rf= 50 breaths/min)

Reviewer 4 Report

The manuscript of Wu et al. (entitled Acute hyperglycemia aggravates lung injury via activation of the SGK1-NKCC1 pathway) shows the worsening of acute lung injury (ALI) by hyperglycemia via the SGK1-NKCC1 pathway in an experimentally induced ALI-model in rats.

The subject of this work would be of interest and the present manuscript provides solid data for a better understanding of the pathophysiology of ALI in acute hyperglycemia.

The experimental procedures and the statistics are appropriate.

The results are clearly presented, the conclusion is correct, but there are some points for criticism that must be addressed in a new submission.

Major points:

  1. The effect of hyperglycemia on the expression of SGK1, total NKCC1 and phosphorylated NKCC1 is convincingly demonstrated in Fig.6. In the same figure, however, there is no significant change of the SGK1 expression in ALI as compared to sham (panel A, first and third columns), although there are significant changes in respect of the NKCC1 expression. How do the authors interprete these findings regarding the pathophysiology of ALI? In addition, at least one block of representative images of immunoblots must be shown, and not only the densitometric analyses of them (Fig.6).
  2. The figure legends of Figs. 3-4-5-6 are extremely long, containing lots of results, which are repeated in the text-body again. The authors must eliminate useless repetitions.

Minor point:

  1. The title of the Fig.6 indicates data of pro-inflammatory cytokines (likely a typing error), but there are no cytokine-results in this figure at all.

Author Response

Comments and Suggestions for Authors

The manuscript of Wu et al. (entitled Acute hyperglycemia aggravates lung injury via activation of the SGK1-NKCC1 pathway) shows the worsening of acute lung injury (ALI) by hyperglycemia via the SGK1-NKCC1 pathway in an experimentally induced ALI-model in rats.

The subject of this work would be of interest and the present manuscript provides solid data for a better understanding of the pathophysiology of ALI in acute hyperglycemia.

Answer: Thank you for your comments. This is the first study that addressed the influence of acute hyperglycemia in NKCC1 in ALI.

The experimental procedures and the statistics are appropriate.

Answer: Thank you for your comments. We did our best to design the study and perform the experiment.

The results are clearly presented, the conclusion is correct, but there are some points for criticism that must be addressed in a new submission.

Answer: We did our best to response the comments. We performed more experiments and addressed a lot of new findings in the revised manuscript.

Major points:

The effect of hyperglycemia on the expression of SGK1, total NKCC1 and phosphorylated NKCC1 is convincingly demonstrated in Fig.6. In the same figure, however, there is no significant change of the SGK1 expression in ALI as compared to sham (panel A, first and third columns), although there are significant changes in respect of the NKCC1 expression. How do the authors interprete these findings regarding the pathophysiology of ALI? In addition, at least one block of representative images of immunoblots must be shown, and not only the densitometric analyses of them (Fig.6).

Answer:

  1. We thank you for your valuable comments. We agree that the previous data cannot fully explain the changes in NKCC1 in lung injury with or without hyperglycemia. During ALI, the WNK4-SPAK-NKCC1 pathway is activated. However, hyperglycemia further activates the SGK1-NKCC1 pathway and exacerbates lung injury. We addressed this issue in the revised manuscript. We provided more information about mechanism of hyperglycemia related aggravation of lung injury in Fig. 8.
  2. We further provided the images of immunoblots as your valuable suggestions.

The figure legends of Figs. 3-4-5-6 are extremely long, containing lots of results, which are repeated in the text-body again. The authors must eliminate useless repetitions.

Answer: We shorten the descriptions in the figure legends as your suggestions.

Minor point:

The title of the Fig.6 indicates data of pro-inflammatory cytokines (likely a typing error), but there are no cytokine-results in this figure at all.

Answer: We are sorry for the careless mistakes. We corrected this mistake. Besides, we provided the data of anti-inflammatory cytokines.

Round 2

Reviewer 1 Report

All the comments have been satisfactory addressed by the authors. The manuscript has been significantly improved and additional data have been included. I have just one minor point:

Line 217, “rate” should be “rare”

Reviewer 3 Report

The substantive changed made by the authors have significantly improved the paper.